# Evaluating access to health and care services during lockdown by the COVID-19 survey in five UK national longitudinal studies

Constantin-Cristian Topriceanu ![ORCID],[1,2] Andrew Wong,[2] James C Moon,[3,4] Alun D Hughes,[1,2] David Bann,[5] Nishi Chaturvedi,[1,2] Praveetha Patalay,[2,5] Gabriella Conti,[6] Gaby Captur[1,2,7]

For numbered affiliations see end of article.

**Correspondence to**
Dr Gaby Captur;
gabriella.captur@ucl.ac.uk

## ABSTRACT

**Objective**  Access to health services and adequate care is influenced by sex, ethnicity, socioeconomic position (SEP) and the burden of comorbidities. Our study aimed to assess whether the COVID-19 pandemic further deepened these already existing health inequalities.

**Design**  Cross-sectional study.

**Setting**  Data were collected from five longitudinal age-homogenous British cohorts (born in 2000-2002, 1989-1990, 1970, 1958 and 1946).

**Participants**  A web survey was sent to the cohorts. Anybody who responded to the survey was included, resulting in 14 891 eligible participants.

**Main outcomes measured**  The survey provided data on cancelled surgical or medical appointments, and the number of care hours received in a week during the first UK COVID-19 national lockdown.

**Interventions**  Using binary or ordered logistic regression, we evaluated whether these outcomes differed by sex, ethnicity, SEP and having a chronic illness. Adjustment was made for study design, non-response weights, psychological distress, presence of children or adolescents in the household, COVID-19 infection, key worker status, and whether participants had received a shielding letter. Meta-analyses were performed across the cohorts, and meta-regression was used to evaluate the effect of age as a moderator.

**Results**  Women (OR 1.40, 95% CI 1.27 to 1.55) and those with a chronic illness (OR 1.84, 95% CI 1.65 to 2.05) experienced significantly more cancellations during lockdown (all p<0.0001). Ethnic minorities and those with a chronic illness required a higher number of care hours during the lockdown (both OR≈2.00, all p<0.002). SEP was not associated with cancellation or care hours. Age was not independently associated with either outcome in the meta-regression.

**Conclusion**  The UK government's lockdown approach during the COVID-19 pandemic appears to have deepened existing health inequalities, impacting predominantly women, ethnic minorities and those with chronic illnesses. Public health authorities need to implement urgent policies to ensure equitable access to health and care for all in preparation for a fourthwave.

## Strengths and limitations of this study

▶ A strength of the study was the implicit age homogeneity of birth cohort participants. This enabled age matching within each cohort for each data collection sweep.

▶ Combining five cohorts spanning multiple age groups (19, 30, 50, 62 and 74 years old) led to a better understanding of how the COVID-19 pandemic has affected different generations.

▶ The longitudinal nature of the cohorts allowed the derivation of individualised non-response weights for each of the included participants. This enabled us to address non-response bias, rendering our results more generalisable.

▶ As self-reported measures were used, the number of care hours needed were subjected to reporting biases. In addition, single categorical outcome variables do not have the capacity to measure the impact spectrum generated by the cancelled appointments or to fully capture the care needs of the participants.

▶ We binarised the ethnicity variable to enable sufficient sample sizes for comparisons, but this precluded more detailed comparisons between the diverse ethnic groups which exist in the UK. Older cohorts (50, 62 and 74 years old) consisted of almost only white participants, so we were unable to describe findings for older people from ethnic minority groups who might have been most adversely affected by the lockdown. Our younger cohorts (19 and 30 years old) included less than 20% non-white participants, which could result in specious associations. Our analysis did not take into account racism as a structural actor to explain the disparities observed, and further work will be needed to address this.

## INTRODUCTION

On 11 March 2020, the WHO declared the novel SARS-CoV-2 (also known as COVID-19) outbreak a global pandemic. As the UK was facing a surge of new cases, the government imposed national lockdown restrictions on

23 March 2020 in order to limit the spread of the virus. Although the restrictions were gradually relaxed, the most widely accepted end date of the lockdown was considered to be on 4 July 2020, when non-essential businesses such as bars and restaurants opened. Delivery of routine care across the UK National Health Service (NHS) was hampered by the pandemic crisis and the lockdown.

Access to health services and adequate care has been previously shown to be influenced by sex, ethnicity, socioeconomic position (SEP) and the burden of comorbidities.[1 2] However, it is unknown whether access to health and care services during the COVID-19 pandemic differed by these factors, potentially further widening already existing health inequalities.[3] Evidence from previous pandemics suggests this possibility, but data are missing in the context of COVID-19 currently. To answer these questions, a web-based survey was sent to participants in five UK national longitudinal studies, spanning multiple generations; data were collected during the core UK lockdown, between 2 May and 1 June 2020. The survey questions can be accessed online (https://cls.ucl.ac.uk/wp-content/uploads/2020/12/COVID-19-Online-Survey-Questionnaire-Wave-1-April-2020-Version-2.pdf). We investigated the number of participants who had a cancelled surgical or medical appointment, and the number of care hours received for self or other household members over a week during the lockdown. We analysed how these outcomes varied by already established factors contributing to health inequalities. The importance of cancellations stems from the potential consequences of healthcare deprivation, while the number of weekly care hours has been shown to predict admission to long-term care facilities especially in the older population.[4]

## METHODS
### Study design
The five UK national longitudinal studies were the National Study of Health and Development (NSHD),[5] the National Child Development Study (NCDS),[6] the 1970 British Cohort Study (BCS70),[7] Next Steps (NS)[8] and the Millennium Cohort Study (MCS).[9] NSHD participants were born in 1946, NCDC in 1958, BCS70 in 1970 and MCS in 2000–2002, and all participants were followed up from birth (all birth cohorts), while NS is a longitudinal cohort study whose participants, born in 1989–1990, were followed up from adolescence. The cohorts were extensively followed up with periodic assessments which have been described elsewhere. During the UK COVID-19 May 2020 lockdown, an online questionnaire was sent to each participant from each cohort. The questionnaire was designed to explore the physical health, health behaviours, social contact and support, loneliness and mental health, household relationships and care needs, housing situation, employment, finances and benefits and education during the height of the COVID-19 pandemic. The questionnaire's format was mostly multiple choice with either binary (yes/no) or categorical response options. Towards the end of the survey, participants also had the option to enter free text to describe their particular COVID-19 experience. There were no inceptives, just an invite to participate sent by email. Two email reminders were sent to NCDS, BCS70, NS and MCS participants who had not started or who had partially completed the survey. A single email reminder was sent to NSHD participants.

### Patient and public involvement
It was not appropriate or possible to involve the study participants or the public in the design or conduct of our research. However, we plan to disseminate the results to the study participants.

### Outcomes
Individuals experiencing a healthcare-related cancellation in the form of a cancelled surgery, medical procedure or other medical appointment at any time since the beginning of the SARS-CoV-2 pandemic were scored as 1, or 0 otherwise. The number of hours of help received for self or other household member in a typical week during lockdown was recorded in six categories: 0, 1–4, 5–9, 10–19, 20–34 and 35+ hours. The variable encompasses both home healthcare and social help. The care provider could be a family member, a friend, a professional paid carer or a voluntary helper. Thus, we had two outcomes: cancelled surgery, medical procedures or other medical appointments, and the number of hours of help received in a week.

### Exposures
Sex was recoded as 0=male and 1=female, while ethnicity was recoded as 0=non-white and 1=white. As NSHD, NCDS and BCS70 consisted mostly of white participants, ethnicity data were examined only for the NS and MCS cohorts. Highest educational attainment and financial difficulties prior to COVID-19 were used as a proxy for adult SEP. Highest educational attainment was categorised as degree/higher, advanced-level exam/diploma, ordinary-level exam/general certificate of secondary education or none. Financial difficulties before lockdown were self-rated using the following options: managing comfortably, all right, getting by and difficult. As many MCS participants were still undertaking education and were financially dependent on their families, their parents' highest education and financial difficulties were used. Childhood social class was recorded according to the UK Office of Population Censuses and Surveys Registrar General's social class, resulting in six categories: professional, managerial and technical, skilled non-manual, skilled manual, partly skilled or unskilled. Participants were asked to report whether they had a long-standing illness (yes/no). In addition, the nature of each chronic illness was also broadly recorded. Thus, our exposures were sex, ethnicity, SEP and the presence of a chronic illness.

## Covariates

Participants who were clinically extremely vulnerable and at high risk of complications from a potential SARS-CoV-2 infection were sent a letter or text message from the NHS or the chief medical officer advising them to shield, that is, to stay at home except for specific purposes and to avoid contact with people they do not live with, except for specific purposes. Receipt of such a shielding letter was recorded as a binary variable (yes/no). The number of hours of help received for self or another household member in a week before the pandemic was recorded as mentioned earlier.

Women were more likely to care for children aged less than 16 years and were more likely to report psychological distress.[10] Thus, sex differences were further explored using the two covariates. The presence of children aged less than 16 years in the household and the self-reported presence of psychological distress during lockdown were recorded (yes/no). Psychological distress was measured using the 12-Item General Health Questionnaire,[11] defined as 0 if <4, and one if ≥4 for NSHD and NS, and using a shortened 9-item Malaise Inventory defined as 0 if <4, and one if ≥4 for NCDS and BCS70 as previously described.[12] For MCS, the Kessler K6 score was used, defined as 1 if ≥13 and 0 otherwise.[13]

In the literature, non-white ethnicity was associated with key-worker status[14] and COVD-19 infection.[15] Thus, ethnicity differences were further explored using the two covariates. Key worker status was self-rated based on whether the participants' work was classified as critical to the COVID-19 response. COVID-19 infection was recoded as 0=no and 1=yes, based on a positive antigen or antibody test or strong personal suspicion due to symptoms.

## Statistical analysis

Statistical analysis was performed in R V.3.6.0. Frequency distribution of continuous data was assessed visually using histograms. Categorical variables were expressed as counts and percentages for each available category. Within each cohort, childhood SEP, highest educational attainment and financial difficulties were converted into cumulative rank probabilities (ridit scores) to quantify the difference in outcomes comparing the lowest with highest SEP (ie, the relative indices of inequality).[16] Models containing all socioeconomic variables were assessed for multicollinearity via the variance inflation method. As childhood SEP was multicollinear with the other two, had the least amount of missing data and could impact on adult behaviours and health outcomes independently of adult SEP,[17] it was used in the subsequent analysis. However, we additionally report the results from the analyses using SEP based on highest educational attainment and financial difficulties, respectively.

Separate regression models were employed using sex, ethnicity, SEP and presence of chronic illness as predictors of cancelled appointments or number of care hours received during lockdown. Generalised linear models with logit link were employed to predict cancelled appointments, while ordinal logistic regression was used to predict the number of care hours received. The proportional odds assumption for ordinal logistic regression was tested using a Brant test.[18] Weights to account for the stratified survey designs of 1946, 1990 and 2000–2002 cohorts[19] have been previously developed. Logistic regression models predicting each participant's response during the COVID-19 data sweep based on known demographic, socioeconomic, household and individual predictors of non-response at previous data collection points were used to calculate non-response weights.[20 21] In the logistic regression models, missing covariate values were generated using multiple imputation. For each COVID-19 survey respondent, the probability of response was calculated, and non-response weights were derived as the inverse probability of response with further calibration to sum to the total number of respondents in each cohort. The stratified survey design and non-response weights were combined to generate a combined weight.[16] An individualised combined weight was derived for each study respondent (full details available in the Centre for Longitudinal Study COVID-19 Survey User Guide, https://cls.ucl.ac.uk/wp-content/uploads/2020/12/UCL-Cohorts-COVID-19-Survey-user-guide.pdf). Predictors were included sequentially one at a time. Sex analyses were adjusted for these individualised combined weights and for the receipt of a shielding letter. All other analyses were similarly adjusted, but ethnicity analyses were additionally adjusted for sex; SEP analyses were additionally adjusted for sex and ethnicity; and chronic illness analyses were additionally adjusted for sex, ethnicity and SEP.

Gender differences were further evaluated by adjusting for the presence of children aged less than 16 years in the household and for psychological distress during the lockdown. As women are more likely to have a chronic disease, gender differences were also evaluated after adjustment for the presence of a chronic disease.[22] Ethnicity differences were further explored by adjusting for key worker status as ethnic minorities have been reported to be over-represented as key workers in the literature.[14] As other studies have shown greater COVID-19 positivity rates among ethnic minorities,[15] further adjustment for COVID-19 infection was pursued for ethnicity in all analyses.

We also explored whether individuals with multiple comorbidities (defined as two or more) were more likely to experience a cancelled appointment or require a higher number of care hours compared with individuals with a single long-standing illness. Comparisons were made using $\chi^2$ test for the cancelled appointment analyses and Mann-Whitney U test for the care hours analyses.

Cohort-specific analyses were conducted initially. Meta-analyses were then performed across the cohorts, only if there was a significant result in at least one of the cohorts. Heterogeneity was evaluated using the Cochran Q test and Higgins $I^2$ statistic. As smaller samples have more sampling errors in their effect estimate, larger effect size might emerge.[23] Thus, funnel

plot asymmetry was evaluated using the Egger test. Metaregression was conducted with age/cohort as a moderator in order to determine whether it was a source of heterogeneity. As the associations between age and our outcomes were likely to be non-linear based on visual inspection, we performed the metaregression using restricted cubic splines modelling.[24]

We ran sensitivity analyses in which we (1) simulated a complete case analysis through multiple imputation to verify the reliability of the observed sex-related differences as the majority of our respondents were female; using the predictive mean matching method, we have generated five complete data sets[25] and performed a pooled regression; the models were not further adjusted for non-response weights; (2) adjusted the number of care hours received during lockdown analyses for the number of care hours received before the pandemic; (3) explored possible deviations from the proportional odds assumption via multinomial logistic regression with the number of care hours grouped into never (0 hours), low (1–9 hours) and high (10+ hours).

## RESULTS

Overall, 15 291 participants (45% of the combined cohorts' participants) responded to the COVID-19 survey as follows: 1241 out of 1842 (NSHD), 5205 out of 8943 (NCDS), 4247 out of 10 458 (BCS), 1921 out of 9380 (NS) and 2677 out of 9909 (MCS). Being female, having a higher educational attainment, having a higher income and reporting better self-rated health were associated with higher response rates.[19]

Any participant who lacked data for at least one outcome variable was excluded, leaving 14 891 eligible participants who were included in the final analysis (characteristics summarised in table 1). A breakdown of data missingness is presented in online supplemental table S1. Overall, included participants were more likely to be female, over 50 years of age and of a higher educational attainment. Older participants were more likely to have a chronic illness, receive a shielding letter, experience a cancelled appointment and require more care hours during lockdown. The chronic illnesses reported spanned a variety of medical

**Table 1** Characteristics of participants by cohort

| Participant characteristics | Cohort study birth year | | | | |
| | 1946 | 1958 | 1970 | 1989–1990 | 2000–2002 |
|---|---|---|---|---|---|
| Sample size | | | | | |
| Questionnaire respondents (n=15 291) | 1241 | 5205 | 4247 | 1921 | 2677 |
| Included participants (n=14 891) | 1154 | 5119 | 4131 | 1876 | 2609 |
| Age (years) | 74 | 62 | 50 | 30–31 | 19–20 |
| Male (%) | 607 (51.88) | 2432 (47.51) | 1708 (41.40) | 633 (34.09) | 770 (29.51) |
| Non-white ethnicity (%) | N/A | N/A | N/A | 361 (19.27) | 367 (14.17) |
| Childhood SEP I–III (%) | 633 (57.18) | 1897 (43.60) | 1727 (48.08) | 1227 (69.36) | 1755 (79.70) |
| Chronic Illness (%) | 842 (73.02) | 3099 (61.24) | 1955 (48.08) | 715 (39.20) | 830 (33.33) |
| Multimorbidity (%) | 390 (33.33) | 1408 (27.83) | 739 (18.18) | 194 (10.64) | 165 (6.63) |
| Shielding letter (%) | 112 (9.61) | 334 (6.57) | 196 (4.77) | 56 (3.00) | 60 (2.30) |
| Presence of children<16 years (%) | 0 (0.00) | 87 (2.13) | 1660 (41.10) | 462 (25.37) | 15 (0.60) |
| Psychological distress during lockdown (%) | 216 (18.77) | 452 (10.25) | 556 (16.07) | 655 (39.15) | 188 (8.29) |
| Key workers (%) | 9 (0.78) | 938 (18.32) | 1396 (33.79) | 583 (31.08) | 196 (7.51) |
| COVID-19 infection-self-reported or positive test (%) | 27 (2.31) | 296 (5.78) | 379 (9.18) | 197 (10.50) | 158 (6.06) |
| COVID-19 infection-positive test only (%) | 1 (0.09) | 19 (0.37) | 17 (0.41) | 12 (0.64) | 7 (0.27) |
| Outcomes | | | | | |
| Cancelled appointments (%) | 376 (32.58) | 775 (15.17) | 494 (11.97) | 234 (12.47) | 303 (11.61) |
| Care hours during lockdown | | | | | |
| 0 | 1073 | 4651 | 3825 | 1724 | 2552 |
| 1–4 | 61 | 112 | 66 | 47 | 47 |
| 5–9 | 10 | 41 | 36 | 10 | 4 |
| 10–19 | 8 | 42 | 20 | 17 | 3 |
| 20–34 | 5 | 18 | 16 | 4 | 2 |
| 35+ | 13 | 54 | 35 | 10 | 1 |

1946 refers to NSHD; 1958 refers to NCDS; 1970 refers to BCS70; 1989–1990 refers to NS; 2000–2002 refers to MCS.
BCS70, 1970 British Cohort Study; MCS, Millennium Cohort Study; N/A, not available; NCDS, National Child Development Study; NS, Next Steps; NSHD, National Study of Health and Development; SEP, socioeconomic position.

**Table 2** Association of sex, ethnicity, SEP and the presence of chronic illness with cancelled surgery, medical procedures or other medical appointments during lockdown

| Cohort study birth year | Sex* | | Ethnicity† | | SEP‡ | | Chronic illness§ | |
|---|---|---|---|---|---|---|---|---|
| | OR (95% CI) | P value | OR (95% CI) | P value | OR (95% CI) | P value | OR (95% CI) | P value |
| 1946 (n=1170) | 0.97 (0.76 to 1.25) | 0.827 | N/A | N/A | 1.39 (0.90 to 2.16) | 0.138 | 1.74 (1.28 to 2.36) | **0.0004** |
| 1958 (n=5073) | 1.20 (1.03 to 1.40) | **0.021** | N/A | N/A | 1.05 (0.78 to 1.41) | 0.753 | 2.15 (1.76 to 2.62) | **<0.0001** |
| 1970 (n=4099) | 1.83 (1.47 to 2.26) | **<0.0001** | N/A | N/A | 1.05 (0.73 to 1.51) | 0.786 | 1.77 (1.42 to 2.21) | **<0.0001** |
| 1989–1990 (n=1849) | 1.70 (1.23 to 2.35) | **0.001** | 1.25 (0.86 to 2.37) | 0.255 | 1.45 (0.88 to 2.41) | 0.154 | 1.59 (1.18 to 2.13) | **0.002** |
| 2000–2002 (n=2605) | 2.29 (1.65 to 3.19) | **<0.0001** | 1.03 (0.73 to 2.31) | 0.885 | 1.05 (0.66 to 1.67) | 0.836 | 1.71 (1.30 to 2.25) | **0.0001** |

All analyses used generalised linear models with logit link. Significant p values are highlighted in bold.
*Sex was coded as 0=male and 1=female; adjustment was made for survey combined weight and shielding letter.
†Ethnicity was coded as 0=non-white and 1=white; adjustment was made for survey combined weight, shielding letter and sex. Almost all participants in NSHD (1946), NCDS (1958) and BCS (1970) were white, so ethnicity was not examined.
‡SEP was coded using childhood social class from 1=managerial to 6=unskilled, but ridit scores were used in all analyses; adjustment was made for survey combined weight, shielding letter, sex and ethnicity.
§Chronic illness was coded as 0=absent and 1=present; adjustment was made for survey combined weight, shielding letter, sex, ethnicity and SEP.
N/A, not available; NCDS, National Child Development Study; NSHD, National Study of Health and Development; SEP, socioeconomic position.

systems (online supplemental table S2). Across all cohorts, the most prevalent conditions were high blood pressure (2119 participants), recurrent back problems (1884 participants), mental health issues (1708 individuals) and asthma (1703 individuals). Individuals with multiple comorbidities were more likely to experience cancelled surgeries, medical procedures or medical appointments during lockdown and to require more care hours than those with a single chronic condition (online supplemental table S3). In NS, non-white participants were less likely to be key workers (p=0.016), but we found no association between ethnicity and COVID-19 infections (p=0.296). In MCS, there was no association between ethnicity, and neither being a key worker (p=0.647) nor being infected with COVID-19 (p=0.979).

### Cancelled surgery, medical procedures or medical appointments during lockdown

In all cohorts except NSHD, female sex was associated with higher odds (OR range 1.20–2.29, all p<0.021) of cancelled surgery, medical procedures or medical appointments (table 2). Adjusting for the presence of children less than 16 years old (online supplemental table S4) and for the presence of psychological distress during lockdown (online supplemental table S5) attenuated the regression coefficients in most cohorts, but sex differences persisted. All the sex differences persisted after adjusting for the presence of a chronic illness, but most coefficients were attenuated (online supplemental table S6). The meta-analysis revealed a pooled OR of 1.40 (95% CI 1.27 to 1.55) in the absence of funnel plot asymmetry (Egger test, p=0.376; table 3). However, there was considerable heterogeneity between the cohorts ($I^2$=85.78%, p<0.0001). In each of the cohorts and in the meta-analysis, the presence of a chronic illness at baseline was associated with higher odds (pooled OR 1.84, 95% CI 1.65 to 2.05) of experiencing a cancelled event. The meta-analysis revealed no heterogeneity ($I^2$=0.00%, p=0.422) and there was no evidence of funnel plot asymmetry when using the SE as the predictor (Egger test p=0.092). Ethnicity and SEP were not associated with cancellations in any of the cohorts. Age was not significant in the metaregression (online supplemental table S7). A visual representation of the cancelled surgery, medical procedures or medical appointments by sex, ethnicity and the presence of chronic illness across the five UK cohorts is presented in figure 1.

**Table 3** Meta-analysis for the respective association of sex and presence of chronic illness with cancelled surgery, medical procedures or other medical appointments during lockdown

| Predictor | N | Study heterogeneity | | | OR (95% CI) | P value | Egger test P value |
|---|---|---|---|---|---|---|---|
| | | $I^2$ | Q | P value | | | |
| Sex | 14 796 | 85.78% | 28.12 | **<0.0001** | 1.40 (1.27 to 1.55) | **<0.0001** | 0.376 |
| Chronic illness | 12 584 | 0.00% | 3.89 | 0.422 | 1.84 (1.65 to 2.05) | **<0.0001** | 0.092 |

Significant p values are highlighted in bold.

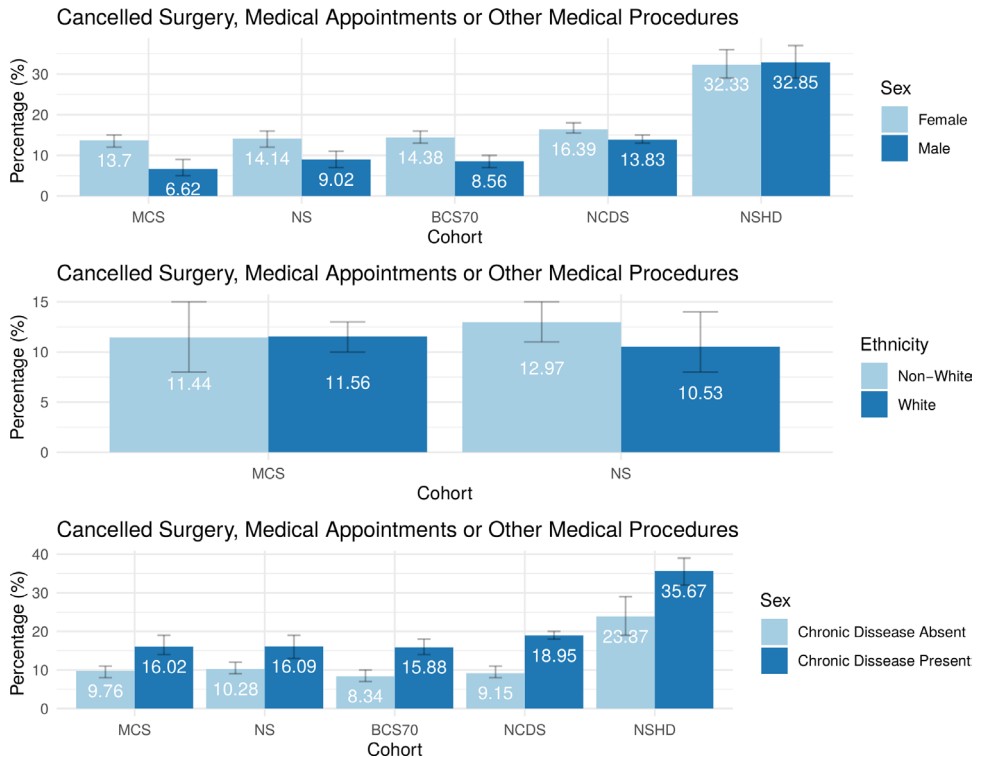

**Figure 1** Bar charts illustrating the percentages of cancelled surgery, medical appointments or other medical procedures by sex, ethnicity and the presence of chronic illness across the five UK longitudinal cohorts, ordered by increasing age of the cohort from left to right. Error bars representing the 95% CIs are also presented. 1946 refers to NSHD; 1958 refers to NCDS; 1970 refers to BCS70; 1989–1990 refers to NS; 2000–2002 refers to MCS. BCS70, 1970 British Cohort Study; MCS, Millennium Cohort Study; NCDS, National Child Development Study; NS, Next Steps; NSHD, National Study of Health and Development.

### Number of care hours for self or another household member during lockdown

In older cohorts, chronic illness was more prevalent, and the association with number of care hours needed was stronger (table 4). In the meta-analysis, a higher number of care hours was associated with ethnic minorities (OR 0.53, 95% CI 0.35 to 0.79, $I^2$=34.17%) and with the presence of chronic illness (OR 2.20, 95% CI 1.72 to 2.56, $I^2$=13·22%; table 5). After adjusting for key worker status, significant associations persisted (online supplemental table S8). After adjusting for COVID-19 infection, significant associations persisted (online supplemental table S9). Sex and SEP were not associated with the number of care hours received during lockdown. There was no evidence that age contributed to the heterogeneity between cohorts from the metaregression (online supplemental table S7). Visual representation of the data is provided in figure 2.

### Sensitivity analysis

Associations between sex and cancelled surgery, medical procedures or other medical appointments persisted after multiple imputation (online supplemental table S10). Adjustment for thr number of care hours received before the pandemic attenuated the ORs, but the associations mostly persisted (online supplemental table S11). Findings were similar in the multinomial logistic regression when looking at the transition from never (0 hours)

to low (1–9 hours), but more variability was observed at the transition from low to high (10+ hours, online supplemental table S12). When using the highest educational attainment or financial difficulties as socio-economical surrogates instead of childhood SEP, there were still no significant associations with cancellations (online supplemental table S13) or the number of care hours during lockdown (online supplemental table S14).

## DISCUSSION

### Statement of principal findings

These data from five UK national longitudinal studies, at the height of the UK national COVID-19 lockdown in May 2020, indicate worrying health inequalities in the access to health and care services—worst hit were women and those with a chronic illness or from ethnic minority groups.

### Meaning of the study

Even before the COVID-19 lockdown, persons with chronic illnesses were vulnerable[26 27] and required more access to health services, as well as care from family members, friends and care service providers.[28] The pandemic triggered unprecedented changes affecting healthcare (which shifted to prioritise COVID-19 patients) and socioeconomic dynamics (caused by restricted movement, changes to work patterns and

**Table 4** Association of sex, ethnicity, SEP and the presence of chronic illness with number of care hours during lockdown

| Cohort study birth year | Sex* | | | Ethnicity† | | | SEP‡ | | | Chronic illness§ | | |
|---|---|---|---|---|---|---|---|---|---|---|---|---|
| | OR (95% CI) | P value | Brant test P value | OR (95% CI) | P value | Brant test P value | OR (95% CI) | P value | Brant test P value | OR (95% CI) | P value | Brant test P value |
| 1946 (n=1170) | 1.17 (0.77 to 1.79) | 0.452 | <0.0001 | N/A | N/A | N/A | 1.17 (0.56 to 2.45) | 0.683 | 0.329 | 2.20 (1.22 to 3.99) | **0.009** | 0.192 |
| 1958 (n=4884) | 1.25 (0.70 to 1.61) | 0.087 | 0.656 | N/A | N/A | N/A | 1.29 (0.81 to 2.06) | 0.282 | 0.877 | 2.17 (1.56 to 3.04) | **<0.0001** | 0.860 |
| 1970 (n=3972) | 0.99 (0.72 to 1.36) | 0.955 | **0.010** | N/A | N/A | N/A | 0.72 (0.39 to 1.30) | 0.272 | 0.026 | 2.74 (1.84 to 4.08) | **<0.0001** | 0.244 |
| 1989–1990 (n=1787) | 0.69 (0.44 to 1.08) | 0.102 | **<0.0001** | 0.44 (0.27 to 0.72) | **0.001** | **0.005** | 0.87 (0.39 to 1.94) | 0.727 | **0.0003** | 1.63 (1.01 to 2.65) | **0.047** | **0.010** |
| 2000–2002 (n=2605) | 0.88 (0.49 to 1.64) | 0.681 | 0.998 | 0.76 (0.38 to 1.53) | 0.432 | 0.972 | 2.17 (0.77 to 6.25) | 0.146 | 0.998 | 1.38 (0.75 to 2.56) | 0.301 | 0.961 |

All analyses used generalised linear models with ordinal logit link. Significant p values are highlighted in bold.
*Sex was coded as 0=male and 1=female; adjustment was made for survey combined weight and shielding letter.
†Ethnicity was coded as 0=non-white and 1=white; adjustment was made for survey combined weight, shielding letter and sex. Almost all participants in NSHD (1946), NCDS (1958) and BCS (1970) were white, so ethnicity was not examined.
‡SEP was coded using childhood social class from 1=managerial to 6=unskilled, but ridit scores were used in all analyses; adjustment was made for survey combined weight, shielding letter, sex and ethnicity.
§Chronic illness was coded as 0=absent and 1=present; adjustment was made for survey combined weight, shielding letter, sex, ethnicity and SEP.
N/A, not available; SEP, socioeconomic position.

remuneration, and unstable housing). Our results show that participants with chronic illnesses were twice as likely to have cancelled medical appointments, potentially depriving them of vital medical care. They were also twice as likely to require increased number of care hours. Only around 50% of the participants had their care hours expectations met, which suggests that a significant proportion were deprived of essential care. Results persisted after adjustment for shielding letter and previous care hours, illustrating their deeply rooted associations with the outcomes. Overall, participants with chronic illnesses received a double hit with potentially long-lasting effects on their health and well-being. These negative effects were even more pronounced for individuals suffering from multiple comorbidities.

We found that women were more likely to experience cancellations in planned surgery, medical procedures or other medical appointments during lockdown. This could be linked to pre-existing sex inequalities where women adopt a more caring role prioritising other family members' needs over their own.[29] Sex inequalities during lockdown could also have widened on account of the added childcare responsibilities, including home schooling, being predominantly undertaken by women. Adjusting for the presence of children under 16 years in the household attenuated the regression coefficients, suggesting this was a likely contributory factor.

Ethnic minorities were twice as likely to require an increased number of care hours compared with white participants in the younger cohorts. It is likely that the unstable socioeconomic landscape dominated by loss of income, unstable housing, increased psychological distress and reduced community support brought about by the lockdown restrictions adversely impacted these communities. Our care hours variable captures both home healthcare and social needs, potentially highlighting broad extra needs during lockdown. Another explanation could stem from the fact that ethnic minorities are over-represented as key workers.[14] To meet the care needs of their communities, they could have been subjected to increased working hours, unusual working environments, stricter work-based controls and greater exposure to COVID-19, exacerbating both physical and psychological stress. However, our data suggest that ethnic minorities were under-represented as key workers in the younger cohorts (NS and MCS).

Although the NHS has an extensive coverage and is free at the point of use, healthcare inequalities have been reported in the UK in the past decade.[30] An important negative finding was the absence of an association between lower SEP and access to health and care services during lockdown. Speculatively, this could mean that the multiple policies implemented by the UK government to address such inequalities have paid off.

Rather surprisingly, the meta-regression showed that age was not a predictor for cancellations or accentuated care needs, suggesting an age-homogenous effect of the lockdown across the generations. We expected that

**Table 5** Meta-analysis for the respective association of sex and presence of chronic illness with number of care hours during lockdown

| Predictor | N | Study heterogeneity | | | OR (95% CI) | P value | Egger test P value |
|---|---|---|---|---|---|---|---|
| | | I² | Q | P value | | | |
| Ethnicity | 4371 | 34.17% | 0.218 | 0.218 | 0.53 (0.35 to 0.79) | **0.002** | N/A* |
| Chronic illness | 12 684 | 13.22% | 4.609 | 0.330 | 2.10 (1.72 to 2.56) | **<0.0001** | 0.312 |

Significant p values are highlighted in bold.
*Egger test was not feasible as only two studies recorded ethnicity.
N/A, not available.

older generations, being more frail and likely to have received a shielding letter than younger persons, would have required more care for activities of daily living and clinical appointments during lockdown. However, the younger generations (NS and MCS) were similarly affected in terms of medical appointment cancellations, as well as the number of care hours required during the pandemic. This is a potentially worrisome indication that the disruption caused by lockdown may have had far-reaching effects on the health and well-being of young people in the UK.

### Implications for clinicians and policymakers

As pandemics can be characterised by multiple waves, they can last several years.[31] Given the prospect of a fourth wave, it is vital that public health authorities implement national interventions to bolster health and care access. In addition, the healthcare disruptions that occurred during the first wave are expected to lead to a surge in late-presenting conditions such as cancer,[32] which will further strain the healthcare system. The challenge facing public health authorities is the need to promote access to healthcare for vulnerable groups on the one hand while minimising infection exposure on the other. Countries without a free healthcare system where citizens rely on paid insurance, such as the USA, are in an even more difficult position.[33]

### Unanswered questions and future research

Remote healthcare known as telehealth, has been brought forward as a potential solution to the problem of health inequalities in the COVID-19 situation. However, telehealth is fraught with similar digital inequalities that will hamper the provision of equitable access.[34 35] To make telehealth egalitarian, factors contributing to digital inequalities need to be addressed. These include technical hardware disparities (lack of technological equipment and slower internet connections), digital literacy and access to technical support.

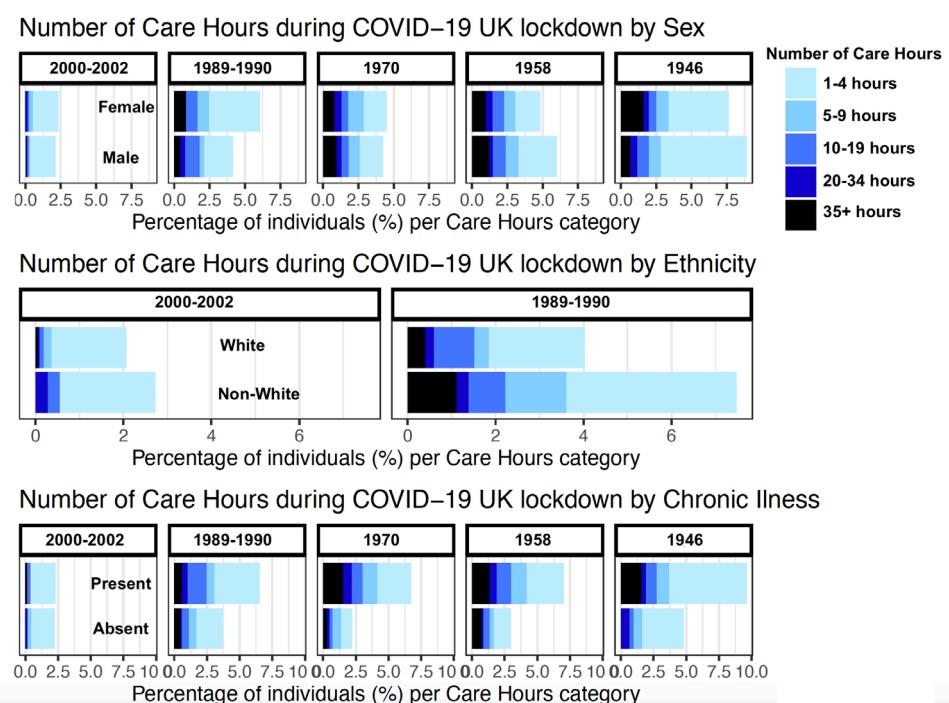

**Figure 2** Bar charts illustrating the percentage of participants requiring support based on the number of care hours needed during the UK COVID-19 national lockdown stratified by sex, ethnicity and the presence of chronic illness across the cohorts.

## Strengths and weaknesses of the study

Strengths of the study are the implicit age homogeneity of participants, enabling age matching within each cohort as participants were exposed to similar life factors before the national lockdown. Combining five cohorts spanning multiple age groups (19, 30, 50, 62 and 74 years old) led to a better understanding of how the COVID-19 pandemic has affected different generations. The longitudinal nature of the cohorts allowed the derivation of individualised non-response weights for each of the participants,[20 21] which have been included in all analyses. This enabled us to address non-response bias, rendering our results more generalisable.

Limitations include data missingness due to low response rates, particularly in younger cohorts, and small sample sizes of older cohorts, particularly NSHD. As self-reported measures were used, the number of care hours needed before and during lockdown were subject to reporting biases. We binarised the ethnicity variable to enable sufficient sample sizes for comparisons, but this precluded more detailed comparisons between the diverse ethnic groups which exist in the UK. Older cohorts (NSHD, NCDS and BCS70) consist of almost only white participants, so we were unable to describe findings for older people from ethnic minority groups that may have been most adversely affected by lockdown. Our younger cohorts (NS and MCS) include less than 20% non-white participants, which could result in specious associations. Our analysis did not take into account racism as a structural actor to explain the disparities observed, and further work will be needed in future studies to address this. By considering chronic illness as a binary variable, we were unable to discriminate between minor and serious illnesses, capture multimorbidity or measure the impact spectrum generated by the cancelled appointments as well as the loss of care hours. In addition, we have not collected any data on the severity of the chronic diseases which could directly influence the need of medical appointments as well as the number of care hours required in a week. We did not capture whether participants had more than one cancelled appointment or procedure. This is especially relevant for participants from lower socioeconomic backgrounds who are more likely to have multiple comorbidities and may have had more than one cancelled appointment or procedure. By reducing many regression variables to a binary coding, we may have underestimated socioeconomic differences which were recorded over a wider categorical spectrum. The survey question about cancelled appointments did not distinguish between face-to-face and virtual clinic consultations. We were unable to separate pandemic effects from recognised confounders such as seasonal variation in the number of care hours needed, as well as other unobserved confounders. The overall prevalence of outcomes differed between cohorts, and this can affect the interpretation of ORs, potentially introducing bias between cohort comparisons. Lastly, a limitation of the restricted cubic spline metaregression is the number of knots per variable as our study included only five cohorts.[36]

## Strength and weakness in relation to other studies

To the best of our knowledge, this is the first UK study to highlight worrying health inequalities in the access to health and care services as a result of the COVID-19 pandemic. In addition, we are the first to address this issue by combining multiple cohorts spanning multiple age groups and with such a high sample size.

## CONCLUSION

Individuals with a chronic illness were more likely to experience cancelled healthcare appointments and greater care needs during the UK national lockdown generated by the COVID-19 pandemic. Women experienced reduced access to healthcare, while ethnic minorities required extra care hours. Our results suggest that the pandemic might have widened pre-existing healthcare inequalities, further depriving already vulnerable and disadvantaged groups of the health and care services which they need. Public health measures should be rapidly implemented to better protect and meet the health and care demands of such at-risk groups ahead of a COVID-19 third wave.

**Author affiliations**

[1]School of Medicine, University College London, London, UK
[2]UCL MRC Unit for Lifelong Health and Ageing, University College London, London, UK
[3]Institute of Cardiovascular Science, University College London, London, UK
[4]Cardiac Imaging Department, Barts Heart Center, London, UK
[5]Center for Longitudinal Studies, Department of Social Science, University College London, London, UK
[6]Department of Economics and UCL Social Research Institute, University College London, London, UK
[7]Center for Inherited Heart Muscle Conditions, Cardiology Department, The Royal Free Hospital, London, UK

**Acknowledgements** The authors thank all the members of our studies for their contribution to this COVID-19 survey and for their ongoing participation in our studies. We thank the survey, data and administrative teams at the Center for Longitudinal Studies and Unit for Lifelong Health and Ageing, UCL, for enabling the rapid COVID-19 data collection to take place.

**Contributors** CCT analysed the data and wrote the manuscript. CCT, JCM, ADH, DB, NC, PP, GCo and GCa were involved in the study design and implementation. AW, ADH, DB, NC, PP and GCo actively participated in the data acquisition. GCa contributed to the data analysis, interpretation of the results and manuscript drafting; was the guarantor of this work; and attested that all listed authors met the authorship criteria and that no others meeting the criteria were omitted. All authors were involved in critically revising the manuscript and approving the final version.

**Funding** The study was funded by the Economic and Social Research Council under the Center for Longitudinal Studies, Resource Center 2015–2020 (grant number ES/M001660/1) and by the Medical Research Council (grant MC_UU_00019/1). GCa is supported by British Heart Foundation (MyoFit46 Special Programme Grant SP/20/2/34841), the National Institute for Health Research Rare Diseases Translational Research Collaboration (NIHR RD-TRC) and by the NIHR UCL Hospitals Biomedical Research Center. JCM is directly and indirectly supported by the UCL Hospitals NIHR BRC and Biomedical Research Unit at Barts Hospital, respectively. DB is supported by the Economic and Social Research Council (grant number ES/M001660/1) and by The Academy of Medical Sciences/Wellcome Trust (Springboard Health of the Public in 2040 award: HOP001/1025). AH receives support from the British Heart Foundation, the Economic and Social Research

Council, the Horizon 2020 Framework Programme of the European Union, the National Institute on Aging, the National Institute for Health Research University College London Hospitals Biomedical Research Center, and the UK Medical Research Council and works in a unit that receives support from the UK Medical Research Council. GCo thanks for support the European Research Council under the European Union's Horizon 2020 research and innovation programme (grant agreement number 819752 DEVORHBIOSHIP–ERC-2018-COG).

**Competing interests** None declared.

**Patient consent for publication** Not required.

**Ethics approval** Ethical approval was obtained from relevant committees and from the University College London/Institute of Education research ethics committee (REC1334). All participants gave informed consent before taking part in the study.

**Provenance and peer review** Not commissioned; externally peer reviewed.

**Data availability statement** Data are available upon reasonable request. NSHD data are available online (https://www.nshd.mrc.ac.uk/data). Data from the remaining cohorts are available from the UK Data Archive (https://www.data-archive.ac.uk).

**ORCID iD**
Constantin-Cristian Topriceanu http://orcid.org/0000-0001-5826-9617

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
