## [Reviewer comments · BMJ Open]

ARTICLE DETAILS

TITLE (PROVISIONAL)	Evaluating access to health and care services during lockdown by the COVID-19 survey in five UK national longitudinal studies
AUTHORS	Topriceanu, Constantin-Cristian; Wong, Andrew; Moon, James; Hughes, A; Bann, David; Chaturvedi, Nishi; Patalay, Praveetha; Conti, Gabriella; Captur, Gaby

VERSION 1 – REVIEW

REVIEWER	Bo Burström Karolinska Institutet, Stockholm Sweden
REVIEW RETURNED	01-Nov-2020

GENERAL COMMENTS	The study concerns an important area – whether the covid-19 pandemic has reduced access to health care, and if so whether there are inequalities in reduced access to care. Data comes from several longitudinal cohort studies, encompassing different age groups from 20-74 years. Two outcome measures were used: cancelled surgery/medical procedures/medical appointment (0/1) and number of hours of help received (6 categories). Socioeconomic position was measured using level of education and financial difficulties. Ethnicity was coded as white/non-white. Chronic illness was classified 0/1. The overall response rate was 45%, higher among females, those with higher educational attainment, higher income and better self-rated health. This may mean that the sample is skewed to represent a more healthy group, and may not adequately represent groups with greater health needs. The authors should discuss how this may affect their results. The predominance of female respondents in the later cohorts also needs commenting – why is that? What are the implications for the findings? Are the findings generalizable? In addition, the measure having chronic illness is a rather wide concept which might include minor (but chronic) ailments as well as more serious and limiting diseases. Was there another question to use to differentiate, or could information from previous cohort data collections be used to measure need of health care? The study is likely to underestimate the presence and importance of comorbidity, and consequently of the need visits. The outcome measures in the study (and some of the exposures) are mainly dichotomous. This reduces the variation, and may have impact on the results. Could this lead to an underestimation of socioeconomic differences? Persons in lower socioeconomic groups are more likely to have multimorbidity, and it is likely that a person with more than one disease may have had more than one cancelled appointment. How would this affect results?
--

	Was there any specification or differentiation regarding the type of cancelled surgery/appointments? Were for instance cancer screening (mammography, Pap smear) more frequently cancelled? Surgery? Were there any digital appointments instead of physical visits? There were no clear socioeconomic differences in terms of cancelled appointments – was this expected, or what are the authors' thoughts about this? What could explain this finding? Some concepts and terms are UK specific and not directly understandable by a non-UK reader. What is, for instance "Shielding letter"? What does it connotate? What does the "number of hours of help received for self or another household member in a week" indicate? Is it social care, or home health care? A short description of responsibilities for health care and social care, and how they overlap, would be clarifying. Ethnic minorities were twice as likely to require an increased number of care hours compared to white participants in the younger cohorts. Is this interpreted as this group not getting ade
--	---

REVIEWER	Jennifer Manuel NYU Silver School of Social Work
REVIEW RETURNED	12-Dec-2020

GENERAL COMMENTS	This manuscript describes an important area of study on access to health services during lockdown of the COVID-19 pandemic. The manuscript is well written and offers important insights. Summarized below are several suggestions for improvement. Methods: More is needed in the Study design section. For example, space permitting, describe the questionnaire in more detail, such as number of questions, length, if there was incentive to participate, and if there were multiple prompts. The timeframe for the outcome, 'cancelled surgery, medical procedures or other medical appointments' is unclear. Was this asked of participants in the last week, some other week during lockdown? Similarly, while the 'number of hours of help received' requests responses for a week during lockdown, it's not clear which week. Also, it would be helpful if the authors clarify the operational definition of number of hours of help received. Was the type of 'help' defined? Did the study include measures on covid testing and positivity rates? While missing data was discussed as a limitation, there is very little written on the extent of missing data in the methods/results sections. I think the finding that "ethnic minorities" were more likely to require an increased number of care hours warrants further discussion. Could this be related to a greater positivity rate among ethnic minorities, as other studies have shown? Additionally, it seems like the authors make a direct causal link between number of clinic cancellations and heightened care needs (measured by increased number of care hours) but without directly testing this relationship (page 9).
---

	Discussion: I'm a little confused by the authors' discussion point on page 11 discussing reduced access to healthcare among ethnic minorities in the US...and then suggesting that similar effects were found in the UK. Clarify how these effects are similar, specifically how reduced access to healthcare in the US study relates to increased number of care hours in this study. The authors also allude to this in the limitations, but the findings are severely limited given that there were no non-white participants in the older cohorts (1946, 1958 and 1970) and very few (19% in 1989-1990 and 15% in 2000-2002). There was also no discussion of how structural factors might play a role in these disparities. It is too simplistic to discuss disparities due to race/ethnicity without discussing racism as a structural factor (see discussion: https://www.healthaffairs.org/doi/10.1377/hblog20200630.939347/full/)
--	---

VERSION 1 – AUTHOR RESPONSE

Reviewer #1- Dr. Bo Burström, Karolinska Institutet

R1–1 The overall response rate was 45%, higher among females, those with higher educational attainment, higher income and better self-rated health. This may mean that the sample is skewed to represent a more healthy group, and may not adequately represent groups with greater health needs. The authors should discuss how this may affect their results. The predominance of female respondents in the later cohorts also needs commenting – why is that? What are the implications for the findings?

Are the findings generalizable?

Thank you for raising this point. As with all surveys, non-response bias can be an issue. In our study, predictors of response were female sex, higher educational attainment, higher income and better self-rated health. However, as these longitudinal cohorts have been extensively followed up, we were able to calculate the probability of response for each participant. Next, non-response weights were calculated as the inverse probability of response and further calibrated to sum to the number of respondents in each cohort. We have now further expanded on the non-response weights methodology and also refer the reader to our publicly available Centre for Longitudinal Study (CLS) COVID-19 Survey Wave-1 User Guide (<https://cls.ucl.ac.uk/wp-content/uploads/2020/12/UCL-Cohorts-COVID-19-Survey-user-guide.pdf>) that extensively discusses on page 28 how the weights were derived and their effectiveness. Therefore, although the response cohort may have been skewed, by adjusting for non-response weights we were able to obtain more generalizable results.

In Methods

“Weights to account for the stratified survey designs of 1946, 1990 and 2000-2002 cohorts¹ have been previously developed. Logistic regression models predicting each participant’s response during the COVID-19 data sweep based on known demographic, socioeconomic, household and individual predictors of non-response at previous data collection points were used to calculate non-response weights^{2,3}. In the logistic regression models, missing covariate values were generated using multiple imputation. For each COVID-19 survey respondent, the probability of response was calculated, and non-response weights were derived as the inverse probability of response with further calibration to sum to the number of respondents in each cohort. The stratified survey design and non-response weights were combined to generate a combined

weight⁴. An individualized combined weight was derived for each study respondent (full details available in the Centre for Longitudinal Study COVID-19 Survey User Guide: <https://cls.ucl.ac.uk/wp-content/uploads/2020/12/UCL-Cohorts-COVID-19-Survey-user-guide.pdf>). Predictors were included sequentially one at a time. Sex analyses were adjusted for these individualized combined weights and for receipt of a shielding letter. All other analyses were similarly adjusted, but ethnicity analyses were additionally adjusted for sex; SEP analyses additionally adjusted for sex and ethnicity; and chronic illness analyses additionally adjusted for sex, ethnicity and SEP.”

In Discussion

“The longitudinal nature of the cohorts enabled the derivation of individualized non-response weights for each of the participants^{2 3} which have been included in all analyses. This enabled us to address non-response bias rendering results more generalizable.”

R1–2 In addition, the measure having chronic illness is a rather wide concept which might include minor (but chronic) ailments as well as more serious and limiting diseases. Was there another question to use to differentiate, or could information from previous cohort data collections be used to measure need of health care? The study is likely to underestimate the presence and importance of comorbidity, and consequently of the need visits.

Thank you for raising this very interesting point. Now we provide a supplementary table with chronic disease breakdown. We provide an additional analysis exploring whether individuals with 2 or more co-morbidities were more likely to experience cancellations or require a higher number of care hours compared to individuals with a single disease.

In Methods

“Participants were asked to report whether they had a long-standing illness (yes/no). In addition, the nature of each chronic illness was also broadly recorded.”

“We also explored whether individuals with multiple co-morbidities (defined as 2 or more) were more likely to experience a cancelled appointment or require a higher number of care hours compared to individuals with a single long-standing illness. Comparisons were made using Chi-squared test for the cancelled appointment analyses and Mann-Whitney U test for the care hours analyses. “

In Results

“The chronic illnesses reported spanned a variety of medical systems (Supplementary Table S2). Across all cohorts, the most prevalent conditions were high blood pressure (2119 participants), recurrent back problems (1884 participants), mental health issues (1708 individuals) and asthma (1703 individuals). Individuals with multiple co-morbidities were more likely to experience cancelled surgery, medical procedures or medical appointments during lockdown and to require more care hours than those with a single chronic condition (Supplementary Table S3).”

In Discussion

“Our results show that persons with chronic illnesses were twice as likely to have cancelled medical appointments potentially depriving them of vital medical care. They were also twice as likely to require increased number of care hours. Only around 50% of the participants had their care hours expectations met which suggests that a significant proportion were deprived of essential care. Results persisted after adjustment for shielding letter and previous care-hours illustrating their deeply rooted associations with the outcomes. Overall, participants with chronic illnesses received a double-hit with potentially long-lasting effects on their health and wellbeing. These negative effects were even more pronounced for individuals suffering with multiple co-morbidities.”

R1–3 The outcome measures in the study (and some of the exposures) are mainly dichotomous. This reduces the variation, and may have impact on the results. Could this lead to an underestimation of

socioeconomic differences? Persons in lower socioeconomic groups are more likely to have multimorbidity, and it is likely that a person with more than one disease may have had more than one cancelled appointment. How would this affect results?

Thank you for raising this. We now better explain the coding of the variables in the manuscript. In addition, we explain here the reasoning behind our chosen outcome and exposure coding. Our outcomes were cancelled surgery, medical procedures or other medical appointments and the number of hours of help received for self or other household member in a week. Cancelled surgery, medical procedures or other medical appointments outcomes was available as a binary variable by study design. Participants were asked to answer the following question "Have you had any surgery, medical procedures or any other medical appointments cancelled since the outbreak of the Coronavirus?" with only Yes/No being available as options. For the question asking about the number of hours of help received respondents could choose from an ordinal set of options broken down as follows: 0, 1-4, 5-9, 10-19, 20-34 or 35+ hours. By using ordinal logistic regression (and multinomial logistic regression as a sensitivity analysis) we were able to explore whether our exposures were associated with an increase in the level of care required. Our exposures were sex, ethnicity, socio-economic position and the presence of a chronic illness. Sex was available as a binary variable. Ethnicity was available only in 1989-1990 and 2000-2002 cohorts (as the other cohorts were almost ethnically homogenous) where it was binary, with all non-White ethnicities aggregated into a single group because their total contribution was only ~15%, so any more granular categorisation would have underpowered our results. Both childhood and adult socio-economic position variables were categorical rather than binary as described in the methodology. Participants were asked whether they had a chronic illness and invited to specify that illness. As we had not collected any data on the severity of the chronic illnesses (a point that we now add to the limitations section) we chose to keep them as a binary variable. Our reasoning was that it was possible for individuals with a solitary severe condition (e.g., cancer) to require more care than those with multiple stable health problems (e.g., high blood pressure, recurrent backache and gastritis etc.). However, following your R1-2 comment, we compared outcomes in individuals with multiple vs. single health problems. The reduced variation brought about by binary variables leading to an underestimation of socio-economic differences is indeed a possibility which we now highlight in the limitations. Similarly, we had not asked whether participants had more than one cancelled appointment, a fact that could also have undermined the socio-economic analyses. This has also been added to the limitations.

In Methods:

"Outcomes

Individuals experiencing a healthcare-related cancellation in the form of a cancelled surgery, medical procedure or other medical appointment at any time since the beginning of the SARS-CoV-2 pandemic were scored as 1, or 0 otherwise. The number of hours of help received for self or other household member in a typical week during lockdown was recorded in six categories: 0, 1-4, 5-9, 10-19, 20-34 or 35+ hours. The variable encompasses both home healthcare and social help. The care provider could be a family member, a friend, a professional paid carer or voluntary helper. Thus, we had two outcomes: cancelled surgery, medical procedures or other medical appointments and the number of hours of help received in a week."

"Exposures

Sex was recoded as 0=male and 1=female, while ethnicity was recoded as 0=non-White and 1=White. As NSHD, NCDS and BCS70 consist mostly of White participants, ethnicity data was examined only for the NS and MCS cohorts. Highest educational attainment and financial difficulties prior to COVID-19 were used as a proxy for adult SEP. Highest educational attainment was categorized as: degree/higher, advanced-level exam/diploma, ordinary-level exam/general certificate of secondary education or none. Financial difficulties before lockdown were self-rated using the following options: managing comfortably, all right, getting by and difficult. As many MCS

participants were still undertaking education and financially dependent on their families, their parents' highest education and financial difficulties were used. Childhood social class has also been recorded according to the UK Office of Population Censuses and Surveys Registrar General's social class resulting in six categories: professional, managerial and technical, skilled non-manual, skilled manual, partly-skilled and unskilled. Participants were asked to report whether they had a long-standing illness (yes/no). In addition, the nature of the chronic illness was also recorded. Thus, our exposures were sex, ethnicity, SEP and the presence of a chronic illness."

In Discussion:

"We binarized the ethnicity variable to enable sufficient sample sizes for comparisons, but this precluded more detailed comparisons between the diverse ethnic groups which exist in the UK. Older cohorts (NSHD, NCDS and BCS70) consist of almost only white participants, so we were unable to describe findings for older persons from minority ethnic groups that may have been most adversely affected by lockdown. Our younger cohorts (NS and MCS) include less than 20% non-White participants, which could result in spurious associations. Our analysis did not take into account racism as a structural actor to explain the disparities observed and further work will be needed in future studies to address this. By considering chronic illness as a binary variable, we were unable to discriminate between minor and serious illnesses, capture multi-morbidity or measure the impact spectrum generated by the cancelled appointments as well as the loss of care hours. In addition, we have not collected any data on the severity of the chronic diseases which could directly influence the need of medical appointments as well as the number of care hours required in a week. We did not capture whether participants had more than one cancelled appointment or procedure. This is especially relevant for participants from lower-socioeconomic backgrounds who are more likely to have multiple co-morbidities and may have had more than one cancelled appointment or procedure. By reducing many regression variables to binary coding, we may have underestimated socio-economic differences which were recorded over a wider categorical spectrum. The survey question about cancelled appointments did not distinguish between face-to-face and virtual clinic consultations."

R1–4 Was there any specification or differentiation regarding the type of cancelled surgery/appointments? Were for instance cancer screening (mammography, Pap smear) more frequently cancelled? Surgery? Were there any digital appointments instead of physical visits? Thank you for raising this point. Only the survey variant sent for the 1946 cohort included a category breakdown based on whether the cancellation was for: a routine out-patient appointment, investigating health problems, a surgical procedure or cancer treatment. For the remaining of the 4 cohorts, participants were only asked this question "Have you had any surgery, medical procedures or any other medical appointments cancelled since the outbreak of the Coronavirus?" with only Yes/No being available as options. Thus, we are unable to provide a more detailed breakdown of which types of appointments got cancelled. The question inquired specifically about cancelled appointments and it did not distinguish between face-to-face and virtual clinic consultations (now added to limitations – see note above).

R1–5 There were no clear socioeconomic differences in terms of cancelled appointments – was this expected, or what are the authors' thoughts about this? What could explain this finding? We agree that this is an important and possibly surprising negative finding which we should have better commented on. In the UK, the National Health Service (NHS) has an extensive coverage including underprivileged areas, being free at the point of use and committed to address healthcare inequalities⁵. The fact that our study did not find any association between access to health and care services during lockdown and socioeconomic position, suggests that the measures implemented by health authorities to protect the economically disadvantaged appear to have paid off.

In Discussion

“Although the NHS has an extensive coverage and is free at the point of use, healthcare inequalities have been reported in the UK in the past decade ⁵. An important negative finding was the absence of an association between lower socio-economic position and access to health and care services during lockdown. Speculatively, this could mean that the multiple policies implemented by the UK government to address such inequalities have paid off. “

R1–6 Some concepts and terms are UK specific and not directly understandable by a non-UK reader. What is, for instance “Shielding letter”? What does it connote? What does the “number of hours of help received for self or another household member in a week” indicate? Is it social care, or home health care? A short description of responsibilities for health care and social care, and how they overlap, would be clarifying. Ethnic minorities were twice as likely to require an increased number of care hours compared to white participants in the younger cohorts. Is this interpreted as this group not getting adequate care from health care services? The part about care hours needs to be further explained – also in terms of the change in hours (good or bad?).

Thank you for requesting these clarifications. We now explain these terms and better discuss ethical differences in the context of increased number of care hours.

In Methods

“Participants that were clinically extremely vulnerable and at high-risk of complications from potential SARS-CoV-2 infection were sent a letter or text message from the National Health Service (NHS) or Chief Medical Officer advising them to shield, that is, to stay at home except for specific purposes and avoid contact with persons they do not live with, except for specific purposes. Receipt of such a shielding letter was recorded as a binary variable (yes/no).”

“The number of hours of help received for self or other household member in a typical week during lockdown was recorded in six categories: 0, 1-4, 5-9, 10-19, 20-34 or 35+ hours. The variable encompasses both home healthcare and social help. The care provider could be a family member, a friend, a professional paid carer or voluntary helper.”

In Discussion

“Ethnic minorities were twice as likely to require an increased number of care hours compared to white participants in the younger cohorts. It is likely that the unstable socio-economic landscape dominated by loss of income, unstable housing, increased psychological distress and reduced community support brought about by the lockdown restrictions adversely impacted these communities. Our care hours variable captures both home healthcare and social needs, potentially highlighting broad extra needs during lockdown.”

Reviewer #2- Prof. Jennifer Manuel, NYU Silver School of Social Work

R2-1 This manuscript describes an important area of study on access to health services during lockdown of the COVID-19 pandemic. The manuscript is well written and offers important insights. Summarized below are several suggestions for improvement.

Thank you very much.

R2-2 Methods: More is needed in the Study design section. For example, space permitting, describe the questionnaire in more detail, such as number of questions, length, if there was incentive to participate, and if there were multiple prompts.

Thank you. We now provide a link to the actual Centre for Longitudinal Study (CLS) COVID-19 Survey Wave-1 document:

<https://cls.ucl.ac.uk/wp-content/uploads/2020/12/COVID-19-Online-Survey-Questionnaire-Wave-1-April-2020-Version-2.pdf>. In addition, we provide a short summary of the survey in the main manuscript and clarify that there was no incentive to participate and the number of prompts sent.

In Introduction

“To answer these questions, a web-based survey was sent to participants in five UK national longitudinal studies, spanning multiple generations; data were collected during the core UK lockdown, between 2nd of May and 1st of June 2020. The survey questions can be accessed via the following link: <https://cls.ucl.ac.uk/wp-content/uploads/2020/12/COVID-19-Online-Survey-Questionnaire-Wave-1-April-2020-Version-2.pdf>.”

In Methods

“During the COVID-19 pandemic (May 2020), an online questionnaire was sent to each participant from each cohort. The questionnaire was designed to explore the physical health, health behaviours, social contact and support, loneliness and mental health, household relationships and care needs, housing situation, employment, finances and benefits and education during the height of the COVID-19 pandemic. The questionnaire format was mostly multiple choice with either binary (yes/no) or categorical response options. Towards the end of the survey, participants also had the option to enter free text to describe their particular COVID-19 experience. There were no incentives, just an invite to participate sent by email. Two email reminders were sent to NCDS, BCS70, NS and MCS participants who had not started, or who had partially completed the survey. A single email reminder was sent to NSHD participants”

R2-3 The timeframe for the outcome, ‘cancelled surgery, medical procedures or other medical appointments’ is unclear. Was this asked of participants in the last week, some other week during lockdown? Similarly, while the ‘number of hours of help received’ requests responses for a week during lockdown, it’s not clear which week. Also, it would be helpful if the authors clarify the operational definition of number of hours of help received. Was the type of ‘help’ defined? Thank you for requesting these clarifications. We now explain the timeframe of our outcomes and provide an operational definition of number of hours of help received as advised.

In Methods

“Outcomes

Individuals experiencing a healthcare-related cancellation in the form of a cancelled surgery, medical procedure or other medical appointment at any time since the beginning of the SARS-CoV-2 pandemic were scored as 1, or 0 otherwise. The number of hours of help received for self or other household member in a typical week during lockdown was recorded in six categories: 0, 1-4, 5-9, 10-19, 20-34 or 35+ hours. The variable encompasses both home healthcare and social help. The care provider could be a family member, a friend, a professional paid carer or voluntary helper. Thus, we had two outcomes: cancelled surgery, medical procedures or other medical appointments and the number of hours of help received in a week”

R2-4 Did the study include measures on covid testing and positivity rates?

Having a COVID-19 test was highly selective in May 2020 being mostly limited to keyworkers and hospitalized patients with symptoms. However, we have added a COVID-19 infection variable which was recorded as 1 for a positive antigen or antibody test or strong personal suspicion due to case-definition symptoms. These data are now presented in Table 1. In addition, we also provide a breakdown of those with the more objective evidence of disease that is a positive antigen or antibody test, regardless of symptoms. Following your R2-5 comment, we have used the COVID-19 infection variable to further explore ethnic differences (see below).

R2-5 While missing data was discussed as a limitation, there is very little written on the extent of missing data in the methods/results sections.

Thank you. We have now added a supplementary table with data missingness breakdown.

In Results

“A breakdown of data missingness is presented in Supplementary Table S1.”

R2-5 I think the finding that “ethnic minorities” were more likely to require an increased number of care hours warrants further discussion. Could this be related to a greater positivity rate among ethnic minorities, as other studies have shown?

Thank you for raising the valid point.

In Methods

“In the literature, non-white ethnicity was associated with key-worker status⁶ and COVID-19 infection⁷. Thus, ethnicity differences were further explored using the two covariates. Keyworker status was self-rated based on whether participants’ work was classified as critical to the COVID-19 response. COVID-19 infection was recoded as 0=no and 1=yes, based on a positive antigen or antibody test or strong personal suspicion due to symptoms.”

“Ethnicity differences were further explored by adjusting for key worker status as ethnic minorities have been reported to be over-represented as key workers in the literature⁶. As other studies have shown greater COVID-19 positivity rates among ethnic minorities⁷, further adjustment for COVID-19 infection was pursued for ethnicity in all analyses.”

In Results

“In NS, non-White participants were less likely to be key workers ($p=0.016$), but we found no association between ethnicity and COVID-19 infections ($p=0.296$). In MCS, there was no association between ethnicity, and neither being a key worker ($p=0.647$), nor COVID-19 infections ($p=0.979$)”

“After adjusting for keyworker status, significant associations persisted (Supplementary Table S8). After adjusting for COVID-19 infection, significant associations persisted (Supplementary Table S9).”

In Discussion

“Although existing literature suggests that ethnic minorities are at increased risk of COVID-19 infection, it wasn’t the case for our cohorts. After adjusting for COVID-19 infection, the associations between non-White ethnicity and requiring a higher number of care hours remained.”

R2-6 Additionally, it seems like the authors make a direct causal link between number of clinic cancellations and heightened care needs (measured by increased number of care hours) but without directly testing this relationship (page 9).

We apologize for this. As stated in our limitations, our study is not able to infer causality. We have now rephrased certain parts to avoid any confusion on this point.

In Discussion

“Rather surprisingly, the meta-regression showed that age was not a predictor for cancellations or accentuated care needs, suggesting an age-homogenous effect of the lockdown across the generations. We expected that older generations being more frail and likely to have received a shielding letter than younger persons, would have required more care for activities of daily-living and clinical appointments during lockdown. However, the younger generations (NS and MCS) were similarly affected in terms of medical appointments cancellations as well as the number of care hours required during the pandemic. This is a potentially worrisome indication that the disruption

caused by lockdown may have had far reaching effects on the health and wellbeing of young people in the UK.”

R2-7 Discussion: I'm a little confused by the authors' discussion point on page 11 discussing reduced access to healthcare among ethnic minorities in the US...and then suggesting that similar effects were found in the UK. Clarify how these effects are similar, specifically how reduced access to healthcare in the US study relates to increased number of care hours in this study. The authors also allude to this in the limitations, but the findings are severely limited given that there were no non-white participants in the older cohorts (1946, 1958 and 1970) and very few (19% in 1989-1990 and 15% in 2000-2002). There was also no discussion of how structural factors might play a role in these disparities. It is too simplistic to discuss disparities due to race/ethnicity without discussing racism as a structural factor.

We apologize for the confusion created. After better reviewing the study cited, we removed that point and rephrased our discussion. Now, there is no attempt to compare the UK to the US situation.

In Discussion

“Ethnic minorities were twice as likely to require an increased number of care hours compared to white participants in the younger cohorts. It is likely that the unstable socio-economic landscape dominated by loss of income, unstable housing, increased psychological distress and reduced community support brought about by the lockdown restrictions adversely impacted these communities. Our care hours variable captures both home healthcare and social needs, potentially highlighting broad extra needs during lockdown. Another explanation could stem from the fact that ethnic minorities are over-represented as key workers⁶. To meet the care needs of their communities, they could have been subjected to increased working hours, unusual working environments, stricter work-based controls, and greater exposure to COVID-19, exacerbating both physical and psychological stress. However, our data suggests that ethnic minorities were under-represented as keyworkers.”

“To the best of our knowledge this is the first UK study to highlight worrying health inequalities in the access to health and care services as a result of the COVID-19 pandemic.”

In addition, we now better explain the limitations of our ethnicity analyses as advised.

In Discussion

“We binarized the ethnicity variable to enable sufficient sample sizes for comparisons, but this precluded more detailed comparisons between the diverse ethnic groups which exist in the UK. Older cohorts (NSHD, NCDS and BCS70) consist of almost only white participants, so we were unable to describe findings for older persons from minority ethnic groups that may have been most adversely affected by lockdown. Our younger cohorts (NS and MCS) include less than 20% non-White participants, which could result in spurious associations. Our analysis did not take into account racism as a structural actor to explain the disparities observed and further work will be needed in future studies to address this.”

REFERENCES:

1. Bann D, Villadsen A, Maddock J, et al. Changes in the behavioural determinants of health during the coronavirus (COVID-19) pandemic: gender, socioeconomic and ethnic inequalities in 5 British cohort studies. medRxiv, 2020.
2. Silverwood RJ, Calderwood, L., Sakshaug, J.W., Ploubidis, G.B. A data driven approach to understanding and handling nonresponse in the Next Steps cohort. London: UCL Centre for Longitudinal Studies: CLS Working Paper 2020/5, 2020.

3. Mostafa T, Narayanan, M., Pongiglione, B., Dodgeon, B., Goodman, A., Silverwood, R.J., and Ploubidis, G.B. Improving the plausibility of the missing at random assumption in the 1958 British birth cohort: A pragmatic data driven approach. CLS Working Paper 2020/6: London: UCL Centre for Longitudinal Studies, 2020.
4. Bann D, Johnson W, Li L, et al. Socioeconomic inequalities in childhood and adolescent body-mass index, weight, and height from 1953 to 2015: an analysis of four longitudinal, observational, British birth cohort studies. *The Lancet Public Health* , 3 (4) (2018) 2018
5. Cookson R, Propper C, Asaria M, et al. Socio-Economic Inequalities in Health Care in England. *Fiscal Studies* , 37 (3-4) pp 371-403 (2016) 2016
6. CDC. Health Equity Considerations and Racial and Ethnic Minority Groups: Centers for Disease Control and Prevention; 2020 [Available from: <https://www.cdc.gov/coronavirus/2019-ncov/community/health-equity/race-ethnicity.html> accessed 21/08/2020 2020.
7. Sze S, Pan D, Nevill CR, et al. Ethnicity and clinical outcomes in COVID-19: A systematic review and meta-analysis. *EClinicalMedicine* 2020;29-30:100630-30. doi: 10.1016/j.eclinm.2020.100630

VERSION 2 – REVIEW

REVIEWER	Bo Burström Karolinska Institutet, Dept of Global Public Health Stockholm Sweden
REVIEW RETURNED	19-Feb-2021

GENERAL COMMENTS	The authors have addressed my comments in a satisfactory way, I have no further comments to make.
---

REVIEWER	Jennifer Manuel NYU United States
REVIEW RETURNED	19-Feb-2021

GENERAL COMMENTS	The authors have adequately addressed the reviewer comments.
--